# Modeling Climate Change Impacts on Rangeland Productivity and Livestock Population Dynamics in Nkayi District, Zimbabwe

**Trinity S. Senda [1,*], Gregory A. Kiker [2,3,*], Patricia Masikati [4], Albert Chirima [5] and Johan van Niekerk [6]**

[1] Department of Land Resource Management and Agricultural Technology, University of Nairobi, Nairobi 00625, Kenya

[2] Department of Agricultural and Biological Engineering, University of Florida, Gainesville, FL 32611, USA

[3] School of Mathematics, Statistics and Computer Science, University of KwaZulu-Natal, Pietermaritzburg 3209, South Africa

[4] World Agroforestry Centre (ICRAF), Lusaka, Zambia; P.Masikati@cgiar.org

[5] National University of Science & Technology, PO Box AC 939, Ascot, Bulawayo, Zimbabwe; albert.chirima@nust.ac.zw

[6] Centre for Sustainable Agriculture, University of the Freestate, Bloemfontein 9300, South Africa; vniekerkJA@ufs.ac.za

\* Correspondence: tssenda@students.uonbi.ac.ke (T.S.S.); gkiker@ufl.edu (G.A.K.)

**Abstract:** Smallholder farmers in semi-arid areas depend on both cropping and livestock as the main sources of livelihoods. Rangeland productivity varies on both spatial and temporal scales and provides the major source of feed for livestock. Rangeland productivity is expected to decline with climate change thereby reducing livestock feed availability and consequently livelihoods that depend on livestock. This study was carried out to assess the impacts of climate change on rangeland productivity and consequently livestock population dynamics using a 30-year simulation modeling approach. The climate scenarios used in the simulations are built from the localized predictions by General Circulation Models (GCMs). The primary climate variables under consideration are rainfall (+/−7% change), carbon dioxide ($CO_2$ up to 650 ppm) and temperature (+4 °C change). This was done by applying the SAVANNA ecosystem model which simulates rangeland processes and demographic responses of herbivores on a temporal and spatial scale using a weekly internal time step and monthly spatial and temporal outputs. The results show that rainfall levels of less than 600 mm/year have the largest negative effect on herbaceous biomass production. The amount of biomass from the woody layer does not change much during the year. The carbon dioxide ($CO_2$) effects are more influential on the tree and shrub layers ($C_3$ plants) than the herbaceous layer ($C_4$ grasses). The $CO_2$ effect was more dominant than the effects of rainfall and temperature. In the baseline simulations, the shrub plant layer increased significantly over 30 years while there is a three-fold increase in the woody plant layer (trees and shrubs) where biomass increased from a 1980 production to that of 2010. The biomass of the herbaceous layer was stable over the historical period (1980 to 2010) with values fluctuating between 200 and 400 $g/m^2$. Grass green biomass has a variable distribution where most production occurred in the fields and cleared areas while lower levels of production were found in the forested areas. The spatial distribution of shrub green biomass was less directly linked to yearly rainfall. Shrub biomass was mostly found in forested areas, and it showed a steady increase in production. Cattle, donkey, and goat populations rose slowly from 1980 but the rise was disrupted by a dry period during the late 1980s to the early 1990s causing a decline in all populations primarily due to grass unavailability. The populations of cattle goats and donkeys started to rise again from 1995 onwards due to improvements in rainfall. Cattle and donkey populations were rising faster than that of goats while sheep population was not changing much for most of the simulation period, otherwise

they declined significantly during the drought of 2002. Similar changes in simulated grass biomass ($g/m^2$) were observed in almost all climate scenarios, except for the peak and low years. The livestock population simulation showed few variations in livestock population under all scenarios. The main conclusion from the study is that $CO_2$ effects on rangeland productivity are much more dominant than the localized effects of rainfall and temperature. This has implications of favoring the growth of the tree and shrub layers over herbaceous layer, which meant that in the long run, the species that are able to use tree and shrub layers may be kept as a livelihood source as they will have a feed source.

**Keywords:** climate change; SAVANNA; simulation; scenarios; livestock; rangeland productivity; $CO_2$ effects

## 1. Introduction

Climate change and variability are some of the most influential factors that impact the livelihoods of livestock farmers in semi-arid areas [1]. This is mainly through their effects on rangeland productivity which plays a critical role as the main feed source for livestock. [2], predicted a 25% loss of livestock production in crop-livestock systems in developing countries as a result of climate change mainly because livestock take time to rebuild after die-offs. In turn, livestock also plays an important role in the livelihoods of farmers by contributing significantly toward food security by alleviating seasonal food shortages through meat and milk and most importantly cash income [3]. However, high mortalities and low productivity were to be the most important constraints in this sector, hence farmers fail to realize the potential benefits from livestock. Poor access to animal health support and technologies as well as dry season feeding shortages contribute immediately to high mortalities [4].

The Intergovernmental Panel on Climate Change reported that the average temperature in Sub Saharan Africa (SSA) is projected to increase between 1.5 °C and 3.1 °C by 2050 [5]. It is also noted that extreme weather events such as droughts, floods, and changes in the frequency and intensity of dry spells are increasing in Southern Africa. The upward trend in temperature is projected to continue beyond 2050 while trends for precipitation vary spatially and from one model to another. The exact nature and extent of the impact of climate change on temperature and precipitation distribution pattern remain uncertain and it is expected to affect the poor and vulnerable societies especially in SSA. Despite the potential to increase agricultural productivity in SSA, climate change remains the greatest challenge. Africa is the most vulnerable region to climate change because widespread poverty limits the people's adaptive capacity [6]. The impacts of climate change on agriculture could seriously worsen the livelihood conditions for the rural poor and increase food insecurity in the region. Climate change is however a gradual process which can be adapted to depending on the resiliency of communities or countries affected.

Carbon dioxide levels in the atmosphere are expected to increase significantly in 2050 [5,7]. This will increase the efficiency of photosynthesis and water use, but also is expected to have greater impacts on $C_3$ plants. High temperatures tend to increase lignification of plant tissues and hence decrease the digestibility of forage. It is also predicted that climate change will induce a shift from $C_3$ to $C_4$ grasses. $C_4$ plants are more efficient in terms of photosynthesis and water use than $C_3$ plants. The $C_3$ forage plants generally have higher nutritive values, but lower yields, while $C_4$ plants contain large amounts of low-quality dry matter and have a higher carbon–nitrogen ratio [8].

Rangelands are under pressure from different drivers such as climate change, degradation, and human activities [9]. The productivity of rangelands in the semi-arid areas varies with time and space with most of them being characterized by grasses, shrubs, forbs, and browse [10]. Rangeland productivity and variability are closely dependent on several ecological conditions such as temperature, rainfall, and soil moisture content. These conditions are known to affect the species composition, abundance, and diversity. Different livestock species have different feeding behaviors and changes in

rangeland condition and productivity may have far reaching consequences on grazing regimes [10–12]. These alterations may have adverse effects on livelihoods that depend on livestock for food and income. The spatial variability of the effects of climate change impacts within rangelands is likely to create inequitable distribution of feed resources [10].

The Malthusian theory of population suggests that the number of individuals (animals) are determined by the resources available creating an equilibrium situation. The concept of equilibrium and non-equilibrium in rangelands has been debated for many years [13]. Some authors argue that ungulate populations are more closely influenced by abiotic factors other than forage availability and density dependence. Boone and Wang [14], however, state that rainfall less than 300 to 400 mm per annum or an annual coefficient of variation of above 30% makes non-equilibrium dynamics more likely. In semi-arid areas, feed constraints are already being felt in terms of scarcity, fluctuating quantity and quality which may even degrade faster in communal areas.

Significant research has been conducted in terms of assessing the vulnerability and adaptation of the smallholder farmers to climate change [15,16]. There is however still much work to be done to improve the understanding of the implications that climate change may have in rangelands and how livestock farmers need to position themselves to hedge against the risks and shocks. There is limited information on rangeland productivity and the extent to which it changes spatially and temporally under different climatic conditions. The spatially explicit nature of rangelands in semi-arid areas together with the temporal variability in rainfall complicates the analysis and planning for sustainable livestock production and development. Because of the varying responses of different areas to climate change there is need for more localized assessments of climate change impacts on rangelands. Farmers need this information so that they can anticipate how the feed base will change and to what extent would it impact their livestock population dynamics under different climatic conditions, which would help them to be able to make informed decisions about sales, feeding and breeding. This paper conducts an initial assessment of the implications of climate change on rangeland condition in terms of feed availability (both quality and quantity) and its variability at both spatial and temporal scales.

This effort is achieved by applying the SAVANNA ecosystem model [17–19] under different climate scenarios within a southern African, communal rangeland landscape.

*Objectives*

- To analyse the trends in biomass (herbaceous, shrub and tree layers) production in relation to rainfall in Nkayi district (Zimbabwe) over a period of 30 years (1980–2010).
- To assess the effect of potential changes in climatic variables (rainfall, temperature, and atmospheric carbon dioxide) on rangeland productivity and livestock population dynamics on a temporal and spatial scale.
- To analyse how the biomass production from the tree, shrub and grass layers vary spatially in different times of the year.

## 2. Methodology

### 2.1. Study Area

The study was carried out in the communal areas of Nkayi district (19° 49′ 59″ S, 28°00′00″ E) of Matabeleland North Province in the central-western part of Zimbabwe (Figure 1). Nkayi district covered an area of 5320 km$^2$ which consist of communal areas and a section of protected forest (Gwampa Forest Reserve). The area has always had crop and livestock farming as the main land uses although there is an expansion of settlements. The agro-ecological conditions in Nkayi are characterized by unreliable rainfalls ranging between 450 and 650 mm/year, and periodic drought spells experienced during the rainy season [4]. The Nkayi district is divided into two zones that are mainly characterized by soil and vegetation types. The area in the southern part and along the Shangani River has the red

sand loams also known as isibomvu/isidaka while the northern part toward Kana river consists mainly of Kalahari sands that are inherently infertile.

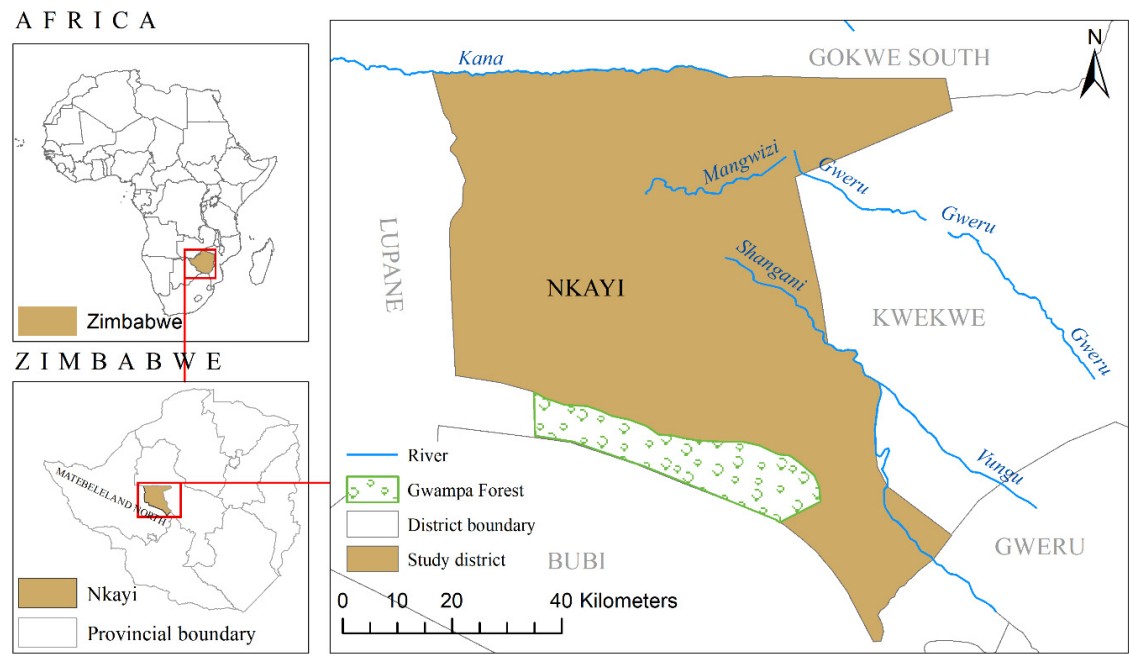

**Figure 1.** Location of Nkayi district in Matebeland north province of Zimbabwe (Data source: International Crop Research Institute for the Semi-Arid Tropics (ICRISAT)).

The vegetation in the red sandy loams is dominated by *Colophospermum mopane*, *Combretum apiculatum*, and various *Acacia* spp. The riverine strips are dominated by different tree species such as *Euclea divinorum*, *Acacia* spp, *Dichrostachys cinerea*, and *Albizia amara*. The northern and greater part of Nkayi District is called sand veld zone and is characterized by diverse miombo woodland dominated by *Julbernadia globiflora* and *Branchstegia Speciformis*. In areas that were previously cultivated regeneration is dominated by *Terminalia serecea* and *Julbernadia globiflora*. Human population densities apart from Nkayi business center ranges from 5–50 pp/km$^2$ with density decreasing from the South to North [3,4].

## 2.2. The SAVANNA Modeling System

This study applies the SAVANNA model (Figure 2) ([17,18,20–22], a grid-based, ecological modeling system. SAVANNA is a spatially explicit and process-oriented model that was developed to address localized management questions for different ecosystem types that include grasslands, shrublands and forests within the savanna biome. The early development and application of the SAVANNA model was in Turkana district of Kenya and improvements to the model were made in subsequent applications [23–25]. The model was designed to address the spatio-temporal variability of ecosystems by using remotely sensed data, GIS databases as well as spatial simulations in order to compute rates of plant production, forage intake by animals and ecosystem functions [26]. It simulates processes in an ecosystem using a weekly time step, simulating vegetation quantity and distribution in response to climate inputs as well as vegetation types consumed by herbivores [19]. The model also simulates the spatial redistribution of herbivores in response to changes in vegetation quantity as well as herbivore demographic responses to changes to biotic and abiotic factors. One advantage of the more meso-scale-focused SAVANNA model to more globalized, savanna ecosystem models such as G-Range [27], aDGVM [28] and LPJ-GUESS [29] is that localized conditions, impacts and potential mitigation alternatives can be simulated and explored in spatial and temporal detail that are meaningful to local managers and stakeholders.

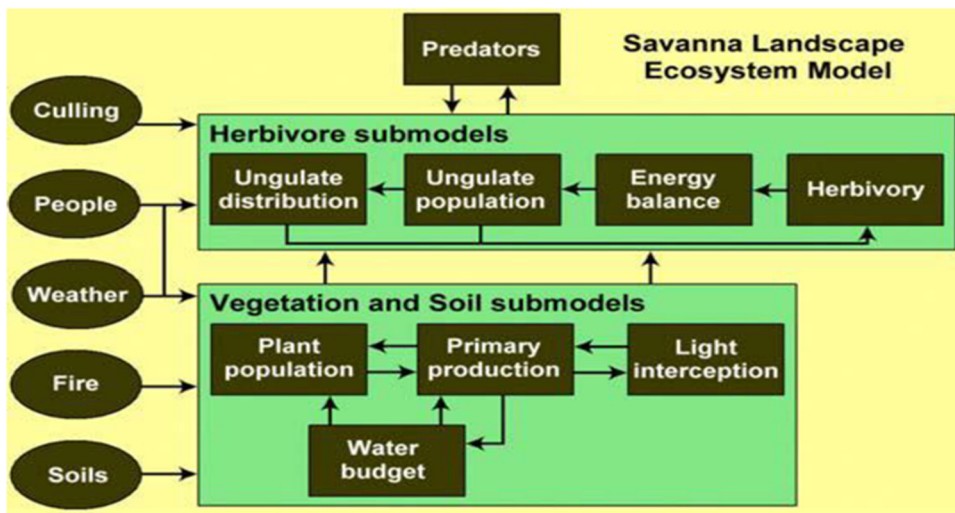

**Figure 2.** Schematic representation of the SAVANNA model [17].

## 2.3. Model Parameterization and Calibration

The SAVANNA model was parameterized to describe the response of changes of different plant and livestock groups. Weather data from 1980 to 2010 were used as the base scenario (Department of Meteorological Services, Zimbabwe) The model parameters used for this study are based on previous SAVANNA research [18,20–22,30] a goat production report by [4] and initial population data collected from Veterinary department in Nkayi [31].

Plants were placed into six functional groups based on previous modeling research [14,18,22,23] to simplify parameterization and to provide computational efficiency when executing simulations. The plant functional groups were based on functional groups described by [32] and classified as follows: Grass ($C_4$ photosynthetic pathway), shrubs ($C_3$ photosynthetic pathway), fine-leaved palatable trees, broad-leaved palatable trees, broad-leaved unpalatable trees and *Colophospermum mopane*. In addition, the grass layer was parameterized with the $C_4$ photosynthetic pathway while all woody shrubs and woody trees were parameterized as $C_3$ photosynthetic pathway species [33]. Livestock groups were also placed into functional groups for similar parameterization and computational efficiency. Herbivore functional groups consisted of cattle sheep, goats, and donkeys. Parameters for herbivore groups were provided by [18] and [34].

As input maps/grids, SAVANNA uses geographic layers (ESRI/ArcMap grids) describing elevation (United States Geological Survey [35] slope, aspect, vegetation, and land use [36], soil class [37], and distance to water to simulate the growth of plants and distribution of livestock. Each grid cell was divided horizontally into one of three vegetation types and vertically into layers of vegetation facets (grass-dominated, shrub-dominated, and tree-dominated) and soils (deep Kalahari sands, Brown loamy sands and Grayish Brown sands). As GIS grids are used as input into SAVANNA, the model outputs both time series files and GIS grids as well. In this study, a 2 km × 2 km grid cell size was used to model the rates of plant production, response of animals to forage availability and ecosystem function under varying climatic conditions. The 2 km grid resolution was selected to provide computational efficiency yet enough detail to simulate the varying vegetation and livestock areas over the entire communal areas of the Nkayi district. The soil input grid was creates using the three soil classes described by [37] (Figure 3) with soil-layer moisture and carbon characteristics derived from [38] and [39].

The vegetation and land use maps were derived from previous land degradation studies [36] and converted to 2 km grid resolution (Figure 4) along with estimates of woody cover and shrub/grass biomass for three primary vegetation types (forest, tilled/fallow field, and degraded). Initial grass biomass amounts were set low to reflect low forage production [36] with 25 g/m$^2$ for fields, 35 g/m$^2$ for conservation forest areas and 10 g/m$^2$ for degraded areas. Woody tree cover in forest vegetation areas was initialized at 70% (20% fine-leafed palatable, 20% broad-leafed palatable, 20% fine-leafed unpalatable and 10% mopane). Woody tree cover in degraded vegetation areas was initialized at 40% (10% fine-leafed palatable, 10% broad-leafed palatable, 10% fine-leafed unpalatable and 10% mopane). Woody tree cover in field areas was initialized at minimal levels (<5%) to represent some presence around human dwellings. Woody shrub biomass levels were initialized at 30 g/m$^2$ and 0.9 m in height for forest and degraded vegetation types and 5 g/m$^2$ and 0.1m for field areas to represent fallow zones and field borders.

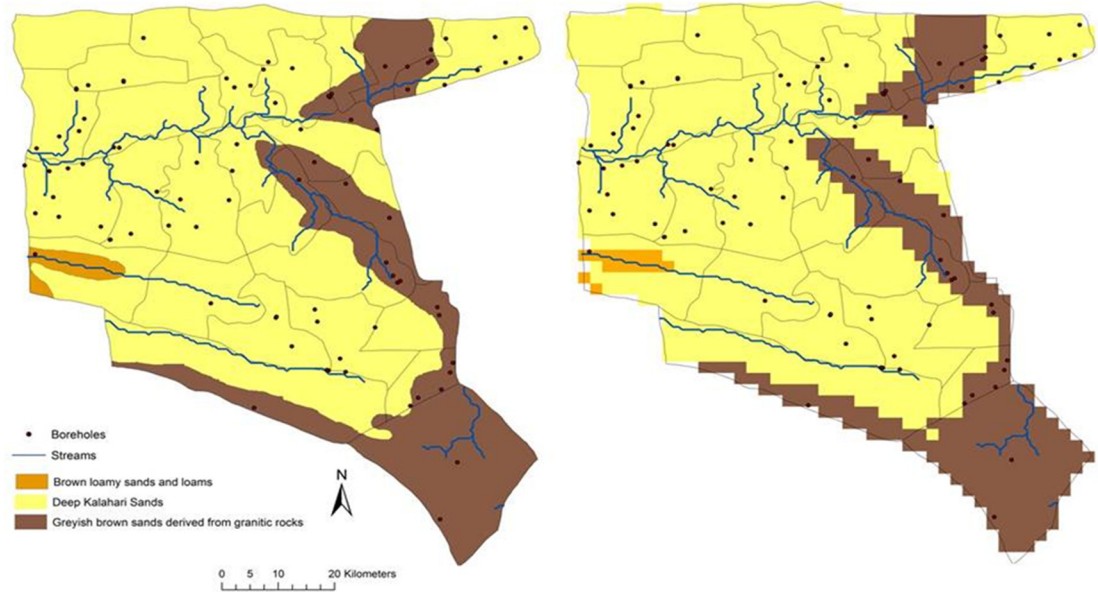

**Figure 3.** Nkayi soils map converted to 2 km resolution grid is used as inputs to the SAVANNA model.

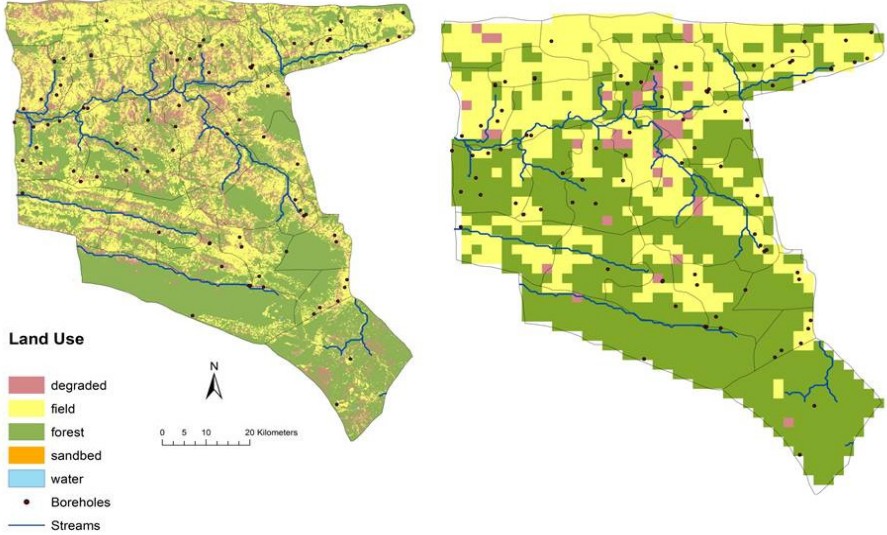

**Figure 4.** Nkayi District land use map converted to 2 km resolution grid used as inputs to the SAVANNA model.

The elevation, slope and aspect were all obtained from the USGS Shuttle Radar Topography Mission (SRTM), 1-arc-second (90 m), digital elevation map [35], that was resampled to 2 km grid resolution (Figure 5). As rainfall in the region has a strong effect on seasonal browse production [10] and livestock performance, two water availability maps were created to represent dry and wet season ephemeral water availability for livestock along with perennially available boreholes. Seasonal water availability (Figure 6) was derived from combining distance to river (perennial and ephemeral) and borehole point data from [36] to achieve the minimum distance (m) for the average wet season (November to April) and the average dry season (May to October).

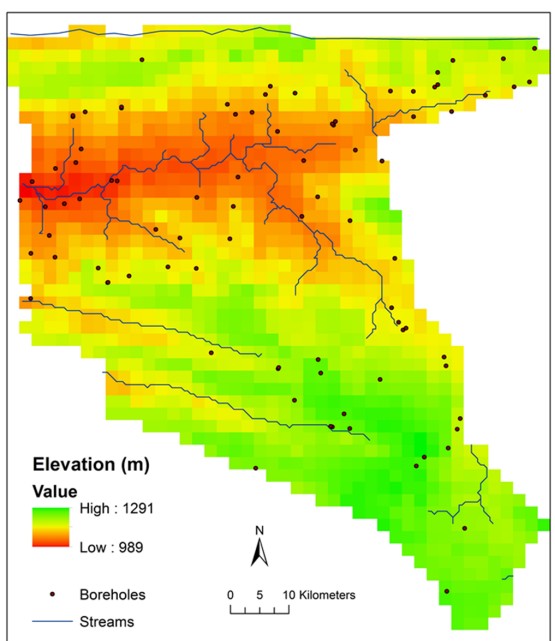

**Figure 5.** Digital elevation map (2 km grid resolution) for the Nkayi district. The grid was resampled from the worldwide SRTM 90 m elevation grid [35].

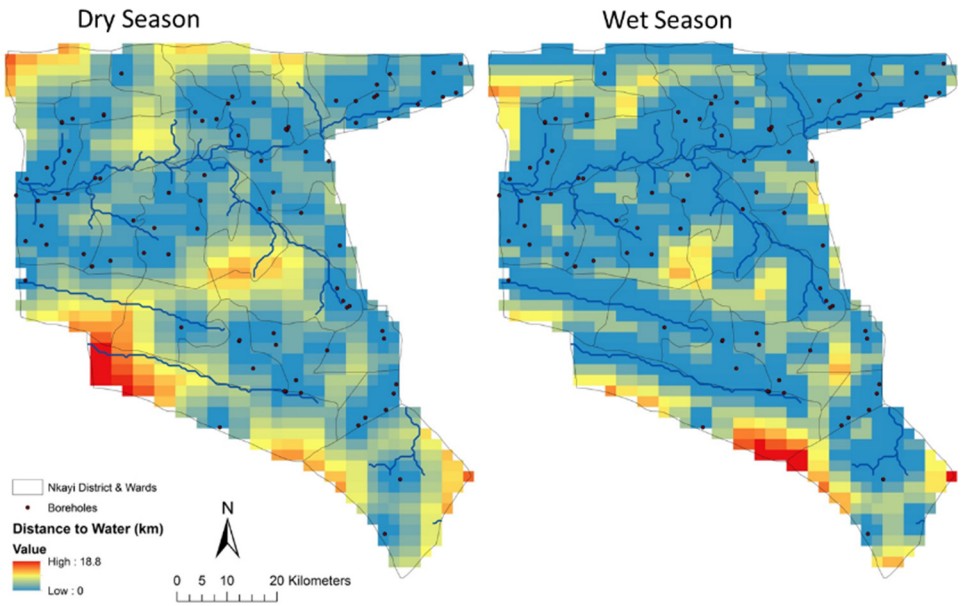

**Figure 6.** Distance to water (km) maps used as inputs to the SAVANNA model. The wet season grid was used in simulations from November to April while the dry season grid was used from May through October.



SAVANNA did not simulate any crop production in the agricultural areas but treated it as available forage for the season. Given that there was significant wildfire suppression and prevention efforts active in the region due to its proximity to forest conservation areas [40], the SAVANNA fire module was switched off (via parameter settings). Thus, any anthropogenic fires were assumed to be of limited size and low intensity as well as fully contained against any accidental spread. As a result, fire dynamics were not simulated in any of the climate scenarios. Additionally, shrub and tree biomass collection for firewood was not present in this version of the model.

Livestock populations in Nkayi district were set to levels described by the Department of Veterinary Services [31] with the total number of cattle at 110,000, one TLU being an animal weighing 350 kg. This level gives stocking rates of approx. 5.0 ha/TLU for cattle. Goat, sheep, and donkey populations stood at 25,000, 15,000, and 10,000, respectively [4]. SAVANNA simulations were executed with two sets of herbivore population demography options to explore the effect of herbivore population on rangeland productivity. The first option was to set all livestock population to constant values based on their initial 1980 levels. The second option allows livestock populations to rise and fall with effects from rangeland condition and water access. In both options, livestock populations were distributed according to habitat suitability with respect to forage, water, and animal densities. This meant that no specific field ownership of specific herds was assigned to control grazing locations as is the norm in most community grazing schemes, so the livestock were generally distributed according to the most satisfactory forage and water suitability. Additionally, no wildlife or other herbivores were simulated in addition to livestock.

## 2.4. Climate Change Scenarios

The Agriculture Model Inter-comparison and Improvement Project (AGMIP), assessed the 20 different GCMs and adopted five models namely CCSM4, GFDL-ESM2M, HadGEM2-ES, MIROC5 and MPIESM [41,42] These five adopted models were used as the basis of this study. These GCMs were chosen as they have a long history in development and evaluation, preference for higher resolution and established performance in monsoon regions ([42]. In the Nkayi district, rainfall was generally predicted to vary by +/−7% while for temperature they are predictions range up to 4 degrees Celsius above historic means. SAVANNA was used to simulate outcomes in the rangeland and livestock numbers from these scenarios. This was done by comparing the current/historic distribution, with simulated climate scenarios. In consultation with Agriculture Model Inter-comparison and Improvement Project (AGMIIP) partners, the different scenarios, and the core rangeland questions for analysis are presented in Table 1. For each climate scenario, the historical thirty-year dataset was altered to uniformly increase the maximum and minimum temperature by an additional 4 degrees Celsius, while rainfall was either increased or decreased by 7% on all monthly inputs. Additionally, each climate scenario was simulated with $CO_2$ levels set at 400 ppm and at 650 ppm to explore the effects of elevated $CO_2$ on the different plant layers. As a result, there were six different scenario simulations for analysis.

**Table 1.** Core questions that guide SAVANNA simulations with historic and GCM Scenarios.

| Core Questions | GCM Scenarios | | | |
|---|---|---|---|---|
| | **Historic Scenario (1980–2010)** | | | |
| | 1. | No change in historical rainfall and temperature | 2. | With/without increased carbon dioxide (400 ppm to 650 ppm) |
| 1. What are the temporal and spatial dynamics of biomass production from the three layers? (tree, shrub, herbaceous) 2. How do the livestock populations respond to changes in the vegetation? | Hot and wet scenario | | | |
| | 3. | 7% increase in rainfall and 4 °C increase in temperature | 4. | With/without increased carbon dioxide (400 ppm to 650 ppm) |
| | Hot and dry scenario | | | |
| | 5. | 7% decrease in rainfall and 4 °C increase in temperature | 6. | With/without increased carbon dioxide (400 ppm to 650 ppm) |

## 3. Results

The results below show the biomass production and livestock population dynamics in the historical and climate scenarios listed in Table 1. SAVANNA simulates biomass quantities at a weekly time step. The green biomass ($g/m^2$) for the herbaceous, shrub and tree layers are used to estimate the peak production period which was selected from the 3rd week of March every year of simulation. This section describes the historical simulation (1980–2010) results as a baseline scenario for subsequent comparison to the subsequent temperature/rainfall/$CO_2$ combinations.

### 3.1. Historical Scenario: Biomass Production over the Entire Nkayi District

The overall Nkayi district average trends in biomass production over the baseline (historical) period of 30 years (1980 to 2010) in relation to rainfall totals are presented in Figure 7.

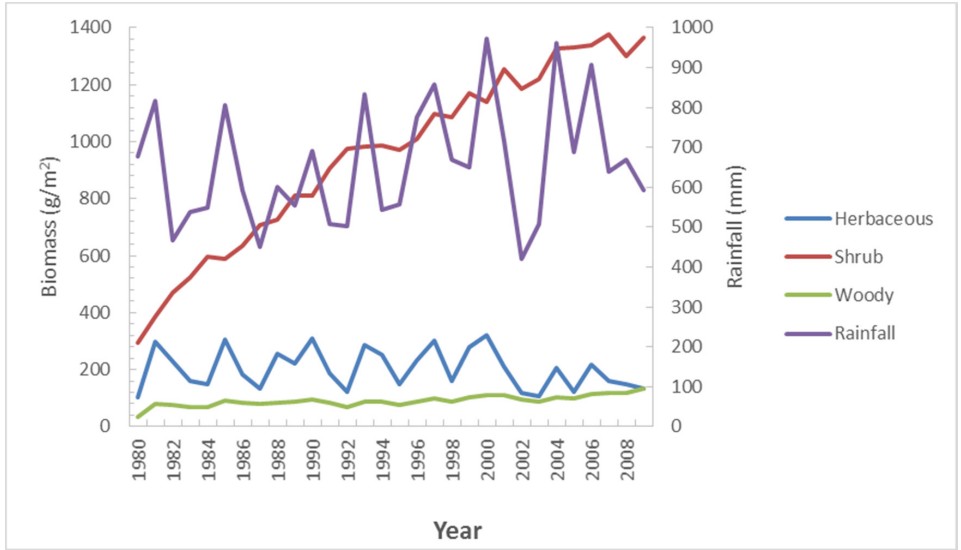

**Figure 7.** Trends in peak biomass production ($g/m^2$) over a period of 30 years (Historic scenario without climate change effects).

In these baseline simulations, the shrub layer increased significantly over 30 years from about 300 $g/m^2$ in the early 1980s to over 1200 $g/m^2$ in the year 2010. A three-fold increase was realized in the woody plant layer (shrubs and trees) where biomass increased from 1980 production levels of approximately 30 $g/m^2$ to being almost steady around approximately 100 $g/m^2$ by 2010. The herbaceous layer does not increase over the historical period with biomass fluctuates between 200 and 400 $g/m^2$ over the years, primarily in response to rainfall amounts. Rainfall levels of less than 600 mm/year have the largest negative effect on herbaceous biomass production. Specific drought years such as 1992 and 2002 have production levels close to 120 $g/m^2$. Conversely, rainfall totals over 1000 mm/year positively influenced biomass production levels to over 300 $g/m^2$. Both the shrub and woody layers have less direct connection to seasonal rainfall totals. A study carried out by [43] on modeling the semi-arid grazing systems showed a pattern similar to the findings of this study on the trends of biomass production. Using the average rainfall and the monthly averages of the biomass yields for the 30-year period a general pattern of the relationship between the rainfall and the biomass was established. The grass layer fluctuates around fluctuating rainfall while the tree and shrub layers are less sensitive to variations. The amount of biomass from the woody layer does not change much during the year. The herbaceous and the shrub layer follow the same pattern, but the shrubs are much higher than the herbaceous layer. The herbaceous layer peaks around February and March and is at its lowest during the August to October period. (Figure 8).

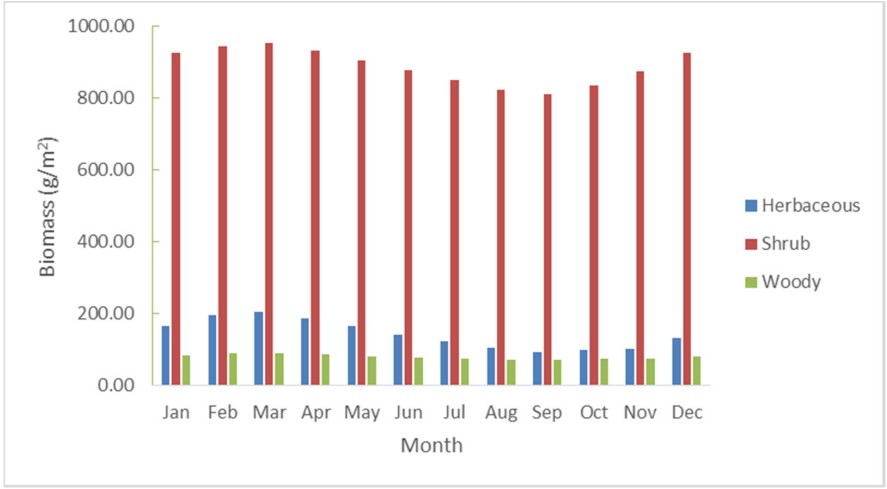

**Figure 8.** The monthly variations in biomass of the three layers during the year (averages for the 30-year period).

### 3.2. Historical Scenario: The Spatial Distribution of Green Biomass Production

Figures 9–11 show how the biomass from the three layers has been distributed in Nkayi over a period of 30 years in the historic scenario. Within Figure 9, grass green biomass has a variable distribution with most production occurring in the fields and cleared areas with chronically lower levels (2–50 $g/m^2$) than found within the forest areas. As was seen in earlier time series graphs, yearly production mostly follows rainfall. Within this simulation, SAVANNA does not simulate crop production areas for part of the seasons so that the availability of forage biomass is probably more limited in reality than shown in these maps in Figures 9–11.

Figure 10 shows the spatial distribution of shrub green biomass which is less directly linked to yearly rainfall. Shrub biomass is mostly found in forest areas, but a significant and steady increase in production is found in all spatial areas over the 30-year simulation. This systematic production increase is mostly likely due to the overall suppression of fire in the system and the exclusive grass consumption of most livestock except goats. In addition, SAVANNA does not simulate any domestic removal of shrub biomass for charcoal, cooking, or other uses.

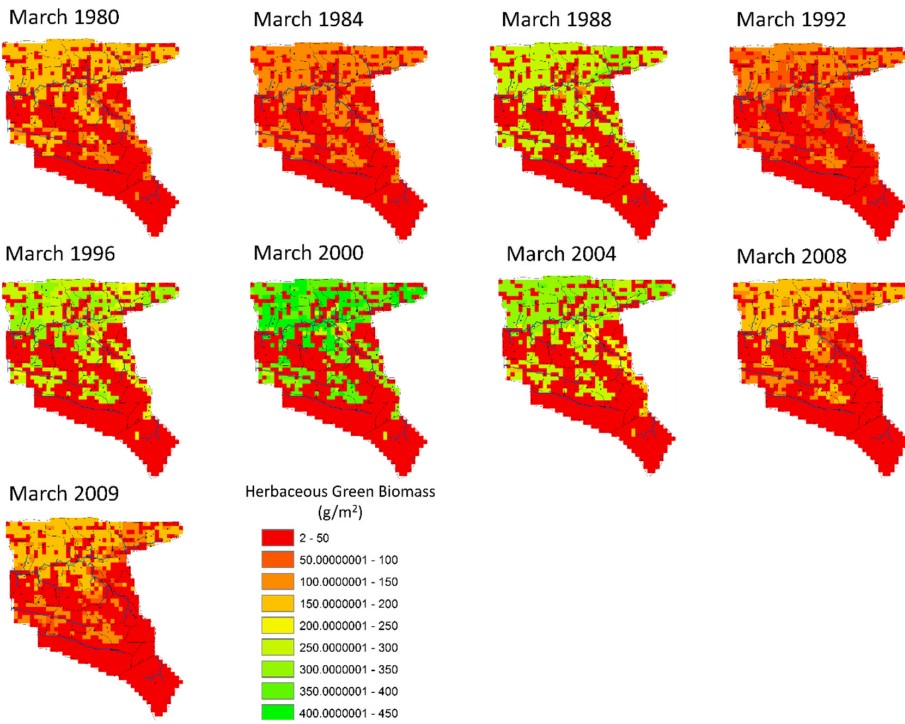

**Figure 9.** The spatial distribution of herbaceous green biomass for selected years (March every 4th year of the historic scenario).

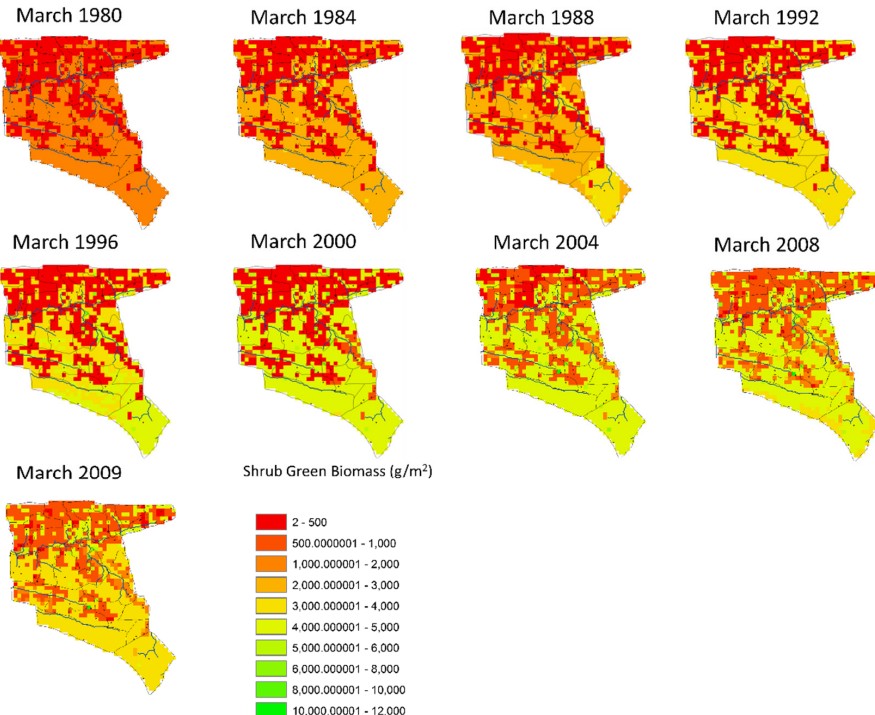

**Figure 10.** The spatial distribution of shrub green biomass for selected years (March every 4th year of the historic scenario).

Figure 11 shows the spatial distribution of tree green biomass which increases at a faster rate across all height classes. The tree green biomass does follow the yearly patterns rainfall of rainfall and also increases systematically over the entire Nkayi area although at different rates according to competition from other plant layers. As with shrubs, only goats will eat the tree biomass in height

classes that are lower than 1.5 m in height. Once tree classes and their biomass have grown past this level, there is no further removal due to fires or domestic human use.

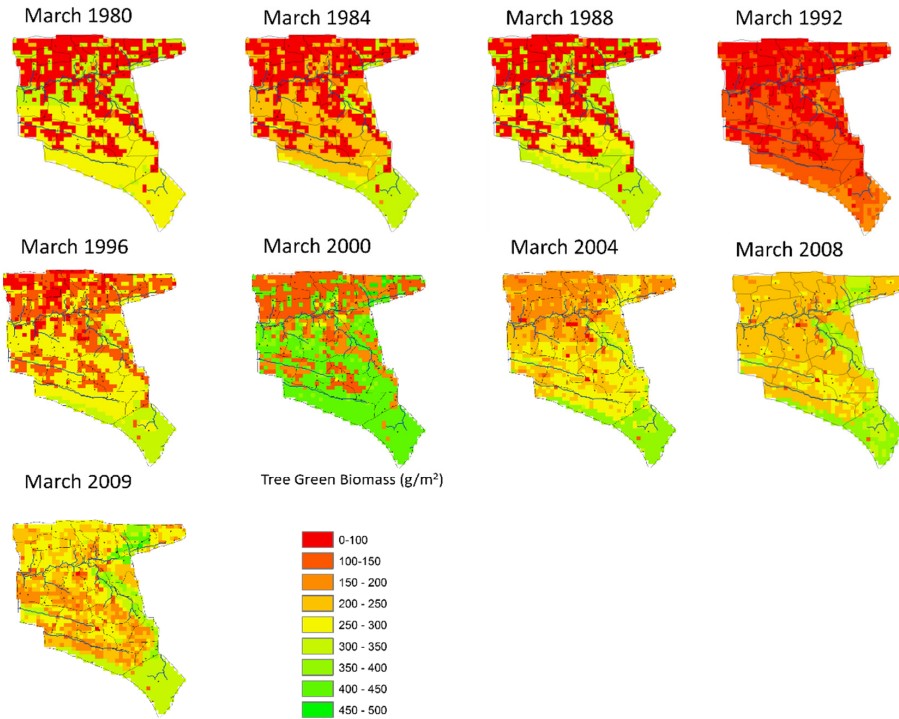

**Figure 11.** The spatial distribution of tree green biomass for selected years (March every 4th year of the historic scenario).

### 3.3. Historical Scenario: The Spatial Distribution of Shrub and Tree Cover

Figure 12 shows a comparison of tree and shrub cover maps for the years starting in 1980, 2000, and 2009. The increase in biomass is shown in Figure 10; Figure 11 above are also realized in systematic and pervasive bush encroachment by mostly woody tree species. Shrub cover does increase as well but in a patchier structure than woody plants. It is suspected that this dynamic is primarily due to the patchy initialization of shrub populations as well as goat browsing of the lower shrub layer and avoidance of the higher woody layers to browsing. Also note that the cover increase is not as linear as seen in the temporal biomass estimates seen in Figure 11. The spatial maps in Figure 12 show relatively slow cover increases in the first 10 to 20 years (1980–2000) and then faster increases over the next decade. This is illustrative of how local management choices can quickly alter the availability and quality of livestock forage.

### 3.4. Historical Scenario: Livestock Population Dynamics

SAVANNA simulations were executed with two sets of herbivore population demography options to explore the effect herbivore population on rangeland productivity. The first option was to set the livestock populations to constant levels' while the second option allowed livestock populations to rise and fall with effects of rangeland condition and water access. In both simulations, the fundamental simulation results in terms were similar that only the fluctuating populations are shown. Figure 13 shows the general livestock population dynamics simulated in most historic and climate change scenarios. Cattle, donkey, and goat populations increase slightly (within the yearly birth/death dynamics) until a dry period from the late 1980s until early 1990s causes a decline in all populations primarily due to grass available. Once higher rainfall amounts arrive in the middle to late 1990s, the three populations rise with cattle and donkey populations rising faster than goat populations. The rapid rise of donkey populations is most likely a function of fecundity/birth parameters and the

lower initial population levels [34] Sheep populations do not change much for most of the simulation and then decline in the significant drought in 2002. This 2002 decline was also realized for goat populations and later for cattle populations. These later populations most likely show that more variable population dynamics are occurring as evidenced by the diminished grass forage resources and increased woody and shrub encroachment. These populations are generally within levels reported by [4] but do not have the detailed harvest dynamics described in these texts. The spatial distribution of livestock for selected years is shown in Figure 14.

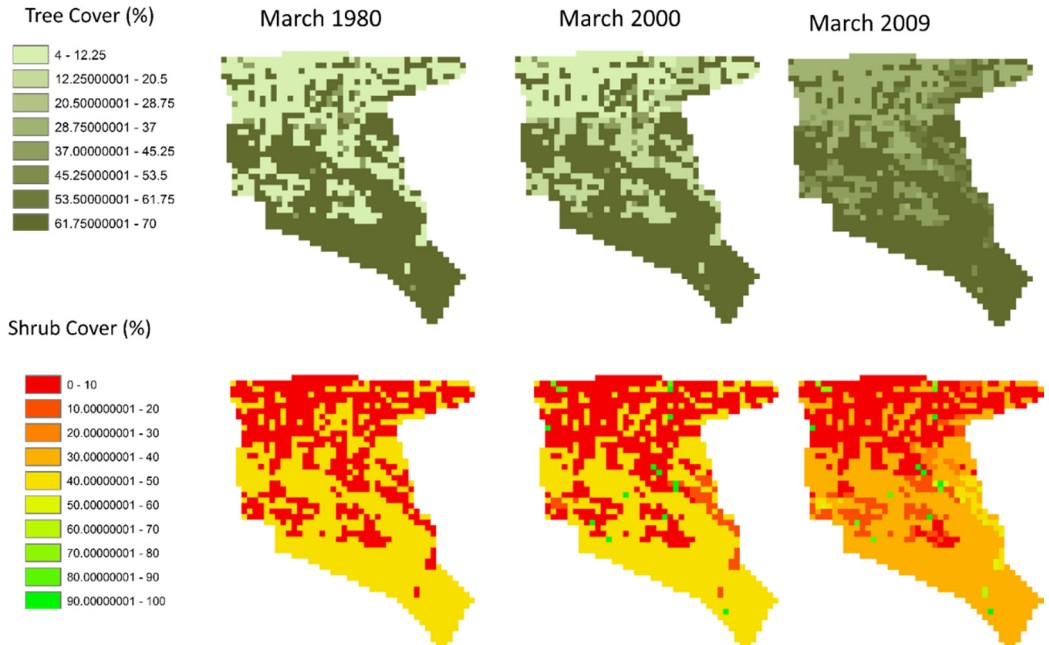

**Figure 12.** A comparison of the spatial distribution of tree and shrub cover for 1980, 2000 and 2009.

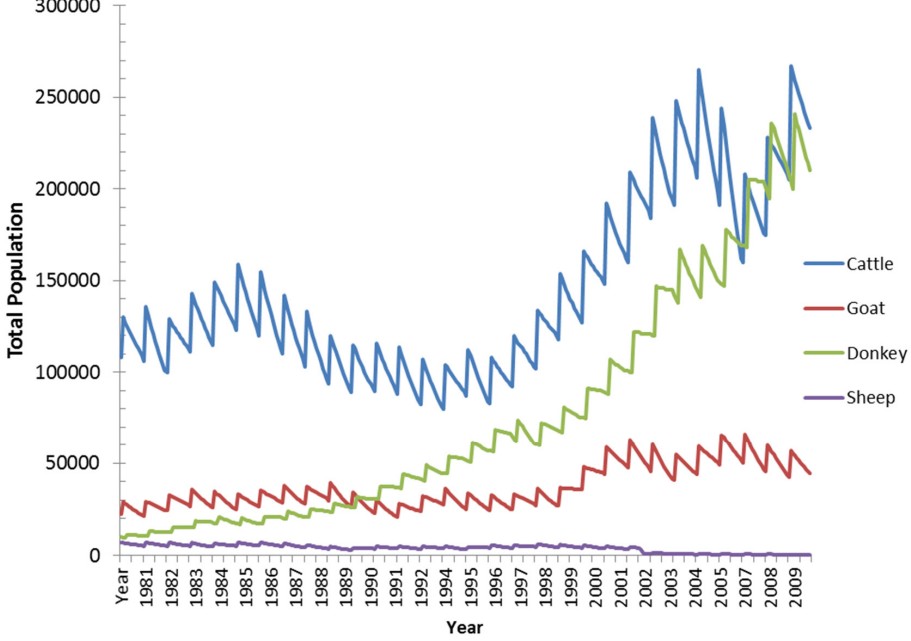

**Figure 13.** Simulated livestock populations for the Nkayi district in the historical scenario.

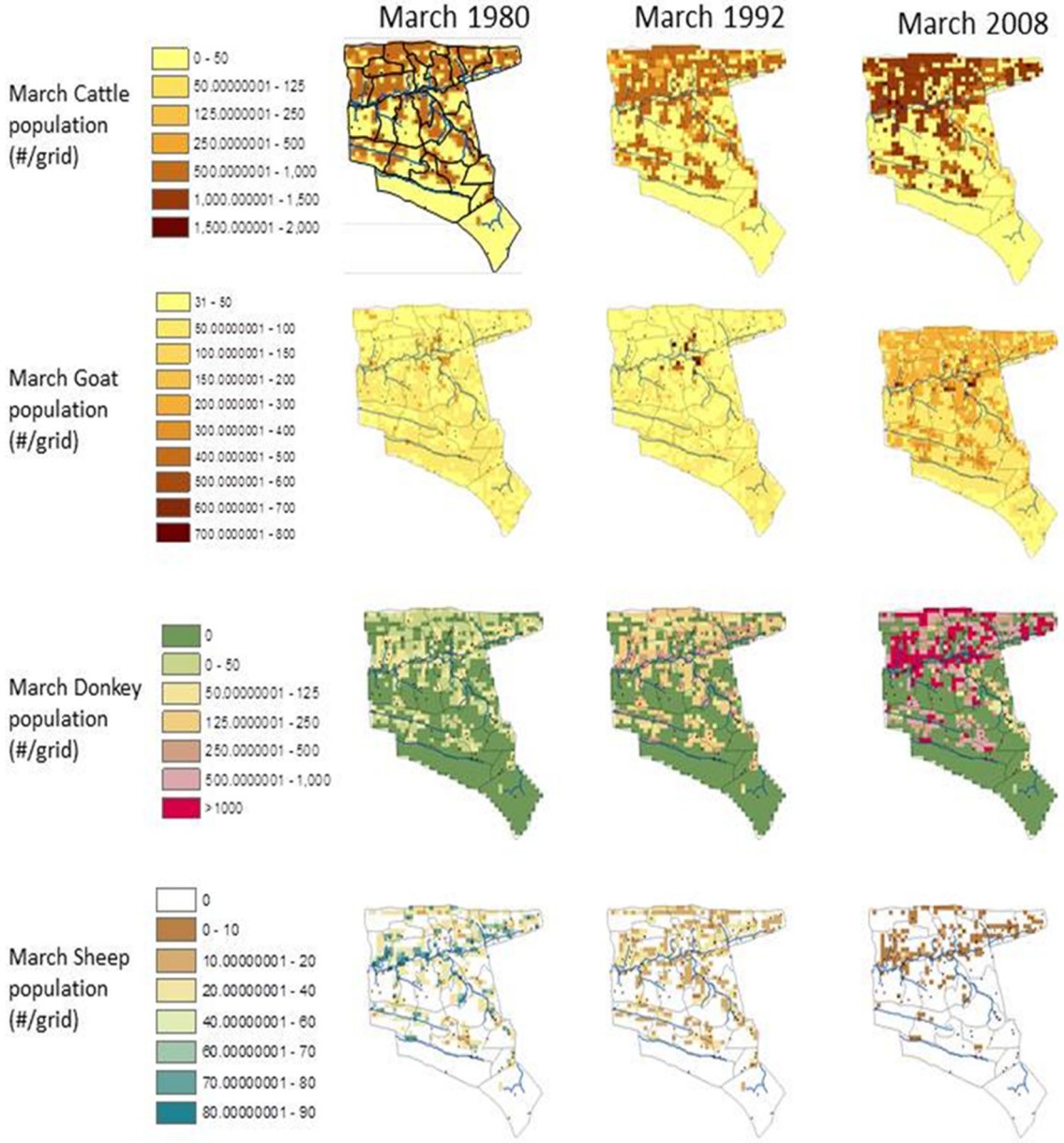

**Figure 14.** Spatial distribution of the livestock for selected years (March 1980, 1992 and 2008).

*3.5. Climate Change Scenarios: Simulated Grass Biomass*

Figure 15 shows simulated grass biomass (g/m$^2$) changing similarly in almost all climate scenarios, except for the peak and low years. These dynamics are mostly similar until about 2003 when the 650 ppm scenarios begin a systematic decrease. This decrease in grass production is probably due to increases in woody competition discussed below. The C4 photosynthetic pathway of the grass layer is generally less responsive to the increases in $CO_2$ levels. This decrease shows a systematic decrease in forage quantity and quality as the effects of livestock population increases and woody encroachment.

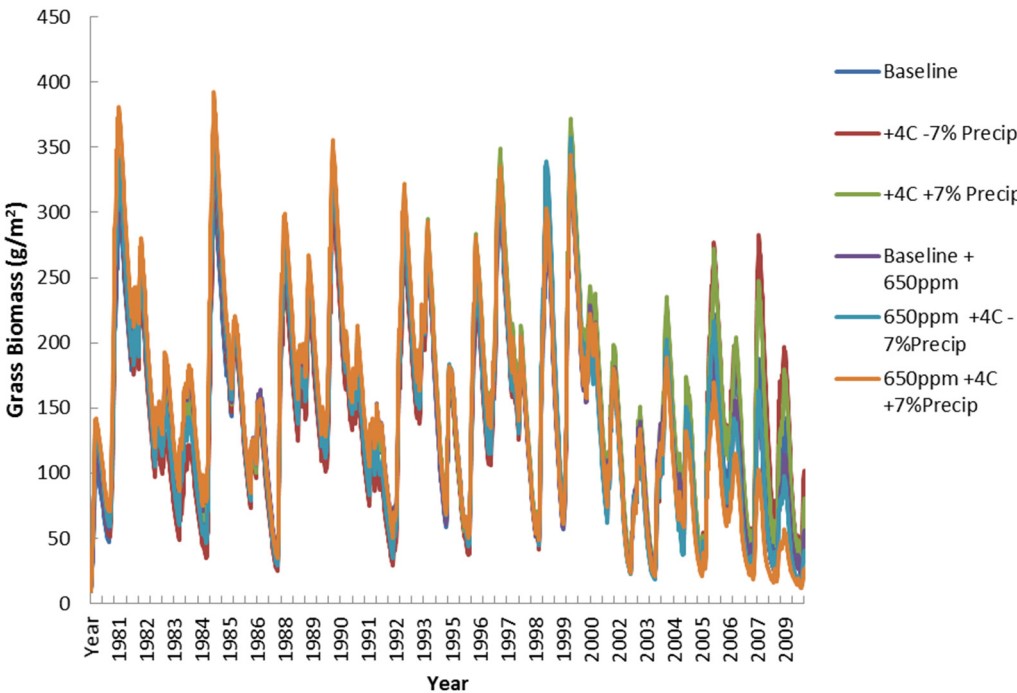

**Figure 15.** Simulated grass biomass (g/m²) for all climate scenarios.

*3.6. Climate Change Scenarios: Simulated Shrub Biomass*

Figure 16 shows a similar systematic increase in shrub biomass as gained by its $C_3$ photosynthetic pathway. The increasing dynamics are similar to some changes in maximum or minimum levels. Shrub biomass levels are somewhat influenced by the $CO_2$ effects, but temperature and rainfall also feature prominently. More variation does occur in the second half of the simulation, but the overall increase is similar among all scenarios. These results have significant implications for the management of the woody shrub layer as grass forage availability may become limiting in later years with increased grazing pressure and changing climate conditions.

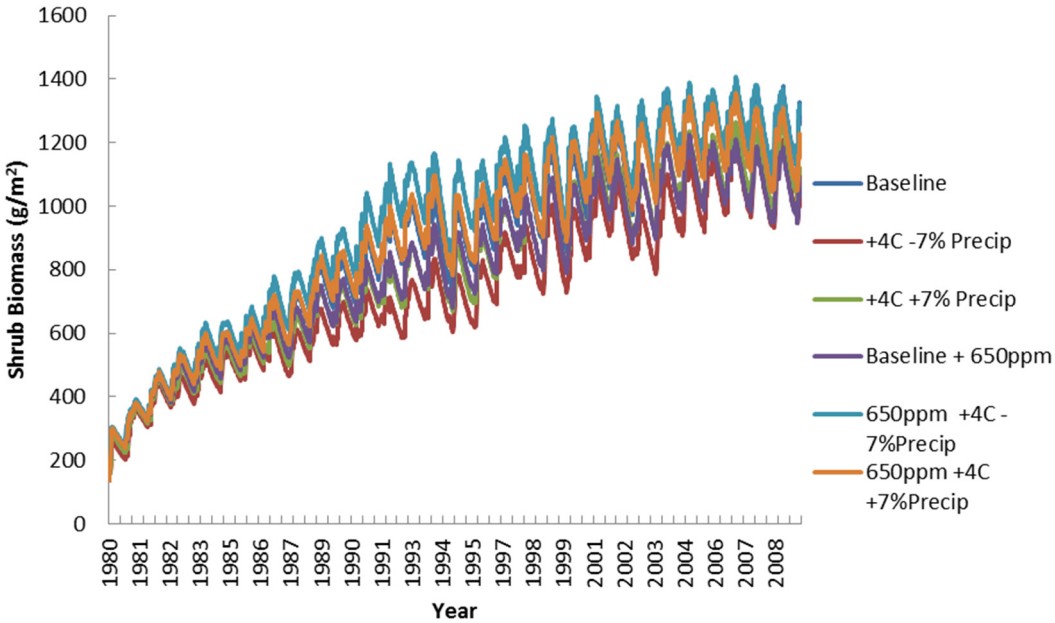

**Figure 16.** Simulated shrub biomass (g/m²) for all climate scenarios.

### 3.7. Climate Change Scenarios: Simulated Tree Biomass

Figure 17 shows the largest differences among scenarios in comparison to the earlier grass and shrub figures. The tree biomass levels are significantly affected by the gains from elevated $CO_2$ and the assumption of a $C_3$ photosynthetic pathway. Interestingly, the three lower levels of woody biomass production are all the scenarios without the 650 ppm $CO_2$ level (Baseline, hot and wet, hot and dry) while the three higher biomass levels of woody biomass are with the 650 ppm of $CO_2$ ($CO_2$ + Baseline, $CO_2$ + hot and wet, $CO_2$ + hot and dry) These results show that the tree layer is more sensitive to elevated $CO_2$ than grass or shrub layers. This $CO_2$ sensitivity is even more important than temperature or rainfall conditions. In addition, the low palatability and height availability of the woody layer would mean lower herbivory and greater opportunity for woody encroachment. Given that fire is already suppressed in these savanna systems, the potential for expansion of the woody layer at the expense of the grass layer is significant.

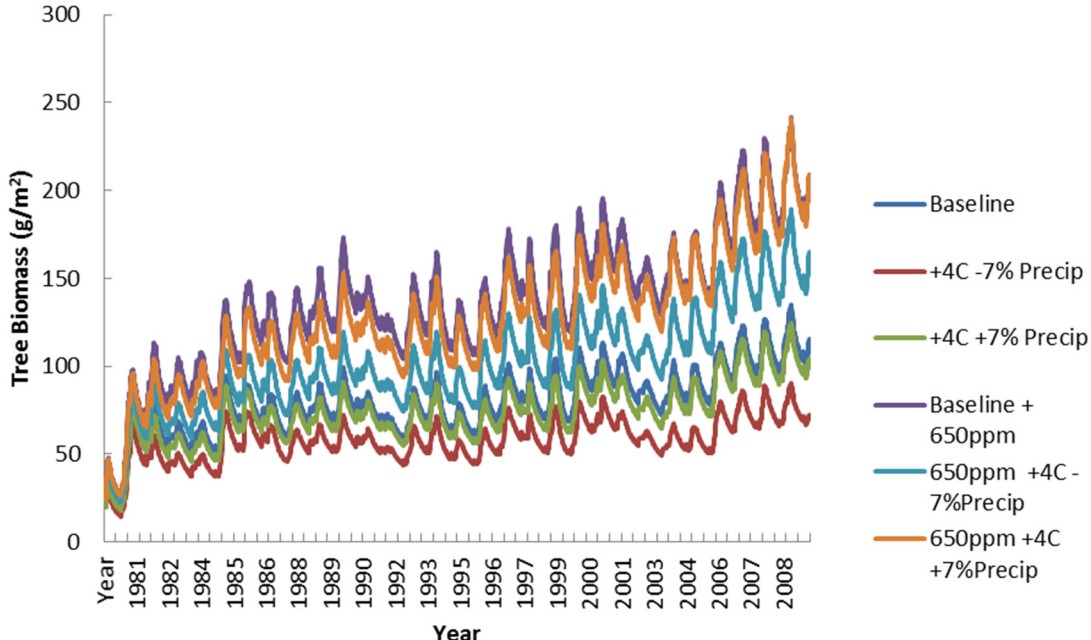

**Figure 17.** Simulated tree biomass (g/m$^2$) for all climate scenarios.

### 3.8. Climate Change Scenarios: Simulated Livestock Dynamics

In all scenario simulations of livestock, populations showed few variations in livestock populations that differed from base line. This trend continued for both fixed and for varying livestock populations. While initial conditions did change the nature of the final populations, additional sensitivity analysis simulations would be useful in determining whether livestock are truly as insensitive to climate conditions or whether additional parameterization is required.

## 4. Discussion

This section summarizes the different SAVANNA simulation results into four general discussion areas. SAVANNA has the advantage of both simulating temporal and spatial dynamics and both elements will be explored in these following sections. Savanna rangelands in Nkayi are characterized by a dynamics and potentially unstable mix of trees and grass growth forms driven by several interacting factors. Tree biomass can increase or decrease producing dense woodlands or open grasslands depending on the prevailing conditions. The trend has been that in many parts of the world there has been an increase in woody plant biomass and decline in open grasslands. The balance between the woody and herbaceous component influences livestock herbivory and an increase in the woody

component is a major concern for livestock farmers. An increase in the woody component generally attributed to climate change and has the effect of reducing the grazing capacity.

### 4.1. Response of Herbaceous Biomass to Yearly Rainfall Totals

Herbaceous biomass responds mostly to yearly rainfall totals and tends to decrease with increase of woody species. This aspect is not surprising as most rangeland research has highlighted the importance of annual precipitation to seasonal herbaceous biomass production [10,11]. In 20 out of the 30-year simulation period, climate change scenarios differed from the baseline only in the magnitude of high and low biomass amounts while the overall trend followed multi-year climate trends. In the last decade of the scenarios, some divergence was seen, as increased woody plant encroachment and a larger herbivore population reduced herbaceous biomass. From a spatial perspective, herbaceous biomass is mostly distributed in the non-forest areas with significantly less production within forest areas with higher woody shrub and tree density as well as completion from these species. These results highlight that the grazing availability and overall quality may be significantly reduced by different combinations of low rainfall, woody plant competition and stocking rates. Additional simulations would aid in disentangling these various environmental and human-induced drivers.

### 4.2. Shrub Biomass and Cover Increase in All Historical and Climate Change Scenarios

This dynamic provides an important result and potential issue for future management consideration. Within the historical scenario, shrub biomass and spatial cover become more dominant over the grass and woody layer toward the mid-1980s to the 1990s, which was also reported in [10,44] The decrease of grass biomass and cover decreased continually as the shrubs increased in the latter part of the thirty-year simulation. This dynamic was quite stable under all climate and $CO_2$ scenario combinations with some small amount variation due to localized conditions. Spatially, the increase occurred in both the open lands as well as forested areas. Localized browsing of goats did have some effects on some grids but overall, more biomass and cover was created than consumed. Some of this increase may be mitigated by higher stocking rates of goats and other human bush reduction activities such as firewood collection.

### 4.3. Woody Biomass and Cover Increase Differently with Respect to Assumptions on Temperature, Precipitation and $CO_2$ Effects

This simulation result was probably the most surprising as woody growth diverged widely within the various climatic and $CO_2$ scenarios. While all woody growth and cover increased from initial levels, some scenarios (hot/dry/400 ppm, hot/wet/400 ppm, baseline/400 ppm) provided less increase than other scenarios (hot/wet/650 ppm, baseline/650 ppm, and hot/dry/650 ppm). In all these comparisons, the level of $CO_2$ was probably the most critical driver in the determination of woody growth. [33] cited that several authors proposed that woody plant expansion is associated with increases in the amount of carbon dioxide in the atmosphere over the past two decades. Other identified impacts of climate change on rangelands are increasing in bush encroachment and alterations in tree-grass interactions. Studies showed that increased carbon dioxide in the atmosphere may cause an increase in the woody layer [45,46]. They also suggest that the rising $CO_2$ levels may reduce the transpiration rates of grasses causing deeper percolation of water which in turn will support the establishment of tree seedlings and enhance growth of many trees. This will in turn increase soil water availability and hence competitive dominance and productivity of deep-rooted plants such as shrubs [33]. The same authors also showed that increased $CO_2$ in the atmosphere favors the post-fire regrowth of woody plants at the expense of grasslands. From the study carried out by Gordon [44], the trees were showed to predominate over the herbaceous biomass leading to bush encroachment. This was attributed it to earlier leaf emergence in trees and greater carbohydrate reserves. For the most part, the SAVANNA results agree with these research findings. Additional sensitivity analysis simulations with various parameter levels would help to explore these effects further.

*4.4. Simulated Livestock Populations Are Not Particularly Sensitive to Climate Change Conditions*

Another surprising result of the SAVANNA simulation is the relative insensitivity of livestock populations to the various climate scenarios. This effect was realized in both stable and set populations of livestock. In all scenarios, grazing herbivores were mostly distributed in the non-forest areas and near to water resources. This spatial limitation may have constrained livestock populations within the limited areas available. Forest areas were not significant resource areas for grazers as their herbaceous biomass was not particularly high, nor were they close enough to surface water to make the areas more favorable for occupation. Goats were the only livestock that were able to use the woodier areas for browsing. These results may be an artifact of model parameterization and further simulations would be warranted before assigning any additional confidence to model results. [10,44] showed that browse is an important feed resource for goats. They also mentioned that if feeding grass and browse become limiting, goats can expand their feed base to counter the effects of low feed quality and quantity. They again showed that cattle are less adaptable to seasonal changes of feed availability. The version of SAVANNA used does not simulate specific livestock management aspects such as moving cattle to specific wet and dry season grazing areas, nor does it simulate the limitation of communal areas due to crop production. Thus, livestock populations were distributed among all suitable areas (mostly in the non-forest areas) without regard to other human land uses. Thus, expanded SAVANNA and human activities simulations such as those by [26] would be useful to add more evidence to any conclusions toward livestock health under climate change conditions.

## 5. Conclusions and Recommendations

At the landscape scale, large herbivore–vegetation interactions can be quite complex involving many interacting factors [19]. Simulation modeling has proved a useful tool for disentangling some of this complexity [19,26]. From this study, trends in biomass production in Nkayi show that the herbaceous layer fluctuates around rainfall, while the woody layer increases at a slow but steady pace. On the other hand, the shrubs increased sharply over the years suggesting that there may be bush encroachment issues on the near horizon. Bush encroachment has a significant effect of reducing the grazing capacity and therefore stocking rates for cattle in Nkayi may need to be adjusted in the long run. Alternatively, farmers could increase off-takes to produce within the acceptable carrying capacity. Bush encroachment can be controlled by using species that are more of browsers than grazers, for example goats. Prescribed burning can also be used to control this potential encroachment. The study also concludes that the forest areas are not a critical feed source for cattle that are mainly grazers as they do not have enough herbaceous biomass (less than 50 g/m$^2$). Additionally, further SAVANNA simulations that include domestic firewood harvesting of shrub and tree layers in both forests and degraded lands would be valuable to include human adaptation mechanisms within this complex system.

It may be concluded from this initial study that climate change effects from the changing rainfall and temperature as predicted by the GCMs may not have strong negative effects on the rangelands near century. However, when considering that the rangelands are currently being systematically degraded from other anthropogenic factors and cannot support sustainable livestock production throughout the year, there is pressing need for a new strategy for agro-pastoral systems to enhance resilience and long-term sustainability.

If carbon dioxide levels continue to increase, the woody layer may respond more strongly than other vegetation layers. As such, there may be a need for farmers to invest in livestock species that are able to use the woody species. The biomass production trends during the year act as a guide for farmers to know when the critical feed shortages are, in terms of biomass production. This can help farmers to plan strategic supplementary feeding so as to avoid loss of body condition by the livestock. It is; however, critical to note that different livestock species have different feeding behaviors and responses to drought. This means that farmers need to plan their management and prioritize the most sensitive livestock species. For example, the 1992 drought had a huge impact on the sheep numbers. This means that those farmers that were using them as the main source of livelihoods were

more adversely affected. Therefore, to hedge against such shocks in future farmers are encouraged to keep a mix if livestock species to potentially spread their risk.

This paper represents an initial simulation effort and not a final one as the results show a complex and coupled human/natural system. Local livelihoods are closely tied with climatic variability and livestock represent a significant and successful human mechanism for mitigating these variations. Whether the grazing system is fundamentally changing beneath resident's feet is a matter for continued discussion. Additional SAVANNA simulations would certainly help to disentangle the various human and environmental drivers to systematically address what human actions (de-stocking, changing from grazers to browsers, water availability) would add to overall resilience of these vulnerable agro-ecosystems.

**Author Contributions:** T.S.S. and G.A.K. both wrote the original manuscript conceptualization and draft. Additionally, they performed model calibration, parameterization and all simulations. P.M. and A.C. aided with model parameterization, and conceptualization as well as reviewing and editing the manuscript. J.v.N. provided review of the manuscript and academic supervision. All authors have read and agreed to the published version of the manuscript.

**Funding:** This research was funded by the Agriculture Model Inter-comparison and Improvement Project (AGMIP), at ICRISAT Bulawayo. Additionally, this work was funded in part by the United States Agency for International Development (USAID) Bureau for Food Security under Agreement # AID-OAA-L-15-00003 as part of Feed the Future Innovation Lab for Livestock Systems. Any opinions, findings, conclusions, or recommendations expressed here are those of the authors alone.

**Acknowledgments:** As authors we are grateful to AGMIP, USAID and ICRISAT for the financial support for undertaking this research. Our sincere gratitude to Dr Michael Coughenour for providing us with many of the livestock parameters as well as providing detailed SAVANNA insights. We are also grateful to the ICRISAT GIS lab for providing us with spatial information.

**Conflicts of Interest:** The authors declare no conflicts of interest.

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
