# Peer review of "Modeling Climate Change Impacts on Rangeland Productivity and Livestock Population Dynamics in Nkayi District, Zimbabwe"

_applsci, doi:10.3390/app10072330_

Round 1

Reviewer 1 Report

This study is a modeling work on simulating the climate change impacts on rangeland productivity. The manuscript is straightforward and well written, while I have a few comments for the authors to improve it.

  1. It is not clear that how and why were the models selected. Are there some other popular models but not used in this study? Please include information on how were these models used in other studies and their efficiency on related objectives.
  2. The citations used in the manuscript are dated. Please include more references from recent years to support the introduction and discussion.
  3. The editing should be carefully checked through the manuscript. Here are some examples but there are more issues to be taken care of:

The font size for the Abstract look inconsistent;

Line 93 add space between two citations;

Line 118 remove the space between “climatic” and the period.

Line 185 remove the extra period at the beginning of the sentence.

Line 439 “carbon dioxide” => “CO2”

  1. In Methodology, what is the land use management history of the studying area?
  2. Figure 8 is the data for the 30 years, or a specific year?

Reviewer 2 Report

Summary In this study, the authors used the SAVANNA model to analyze the impacts of climate change on vegetation biomass (grass, shrub, and trees) and livestock response to the changes under three GCM scenarios in northwestern Zimbabwe. The authors conclude that simulated shifts in rainfall and temperature under different GCMs are unlikely to have serious negative effects on rangelands this century but anthropogenic factors would likely lead to rangeland degradation and reduction in rangeland productivity.   This is a welcome study as it addresses issues of climate vulnerability in a developing country where people are least able to adapt to climate change, and contributes to our understanding of climate change impacts in one of the most important and extensive ecosystem in Africa. The manuscript is well-written but will require some clarifications and elaborations, particularly in the 'Methods' section, and in the assumptions of the simulation model. The datasets and sub-models in SAVANNA should be described more fully and systematically. Also, clarify how were population, fire sub-models were parameterized. One concern I have is the fact that removal of shrub species (an important activity in sub-Saharan Africa) is ignored in the model. This should be discussed in the results as a limitation of the model. Why is fire assumed to be suppressed when several studies on wildfires in southern Africa (e.g., Archibald, 2016; Strydom, 2016) suggest an increase in area burned, as well as the extent of large, intense, and extreme fire events?   Specific Issues Line 88: crop -livestock farmers? Line 118: Remove space between 'climatic' and '.' Line 144: be consistent on whether to abbreviate area as 'km2' or write it out in full - 'square kilometers'. Figure 1. Map needs to be improved to supplement description of study area. Consider showing the two zones described in line 148 along with the Shangani River in the Nkayi District map. The locator (inset) map could be improved to show location of Zimbabwe in southern Africa, and perhaps include only the Nkayi District polygon and one or two major cities in the country for context. Lines 152-159: Please include information sources/citations.  Line 158: Abrupt transition from vegetation to population. Specify you are referring to human population density. Lines 164-165: Paraphrase sentence here as it is taken directly from Coughenour (1993). Line 180-181: Replace 'was' with 'were' since data is plural. Add a full stop after Zimbabwe. Line 185: Delete full stop at beginning of paragraph. Line 188: Replace '.' with ',' after 'pathway)'   Lines 196-206: Please provide additional details on data inputs to the model so that readers can fully understand the characteristics of these datasets without having to go to the cited sources. For example, what time period does the vegetation map capture, and how many vegetation classes are represented? What are the water sources that were used to compute 'distances to water'? Likewise provide some basic details on the soil dataset. Was this a vector source that was rasterized?  Line 230-231: How were the wet and dry season grids created? Were these from precipitation data? If so, the caption under figure 6 (Lines 232-234) is confusing because it refers to a distance to water, there is a need to explain why or how that distance would vary seasonally. How was the 'Fire' sub-model parameterized? Is fire assumed suppressed? If so, what would be the basis for such an assumption? Lines 279-281: '...while the woody and shrub layers...' Are shrub layers not woody too? Line 282: 'shrub biomass'? Line 301: '...exclusive grass consumption of most livestock except goats'? Please clarify. Also clarify about the assumption on fire suppression. Line 302-303: Wouldn't ignoring shrub removal for domestic use not limit the usefulness of this simulation model? Lines 319-320. But shrub is also woody. Lines 375-379: Something is missing in the sentence here.
